# Performance of the Vitek 2 Advanced Expert System (AES) as a Rapid Tool for Reporting Antimicrobial Susceptibility Testing (AST) in *Enterobacterales* from North and Latin America

Cecilia G. Carvalhaes,[a] Dee Shortridge,[a] Leah N. Woosley,[a] Nabina Gurung,[a] Mariana Castanheira[a]

[a]JMI Laboratories, North Liberty, Iowa, USA

**ABSTRACT** This study evaluated the performance of the Vitek 2 Advanced Expert System (AES) confidence level report as a rapid tool for reporting antimicrobial susceptibility testing (AST) results for a challenging set of *Enterobacterales* isolates from North and Latin America. *Enterobacterales* isolates ($n = 513$) were tested by CLSI broth microdilution (BMD) and Vitek 2 (N802 and XN15 AST cards). Wild-type isolates and isolates harboring acquired $\beta$-lactamases by whole-genome sequencing were included. The AES assessment of confidence level (green, yellow, and red reports) was compared to BMD results and known genotypes and reviewed by a microbiologist for accuracy. Totals of 148 (28.8%) wild-type isolates and 365 (71.2%) *Enterobacterales* isolates harboring carbapenemase (211 [41.1%]), extended-spectrum $\beta$-lactamase (ESBL) (122 [23.8%]), and/or transferrable AmpC (tAmpC) (32 [6.2%]) genes were evaluated. The AES confidence level was assessed for 488 isolates, and a phenotype was recognized for 447 (91.6%) isolates. Green, yellow, and red AES reports were noted for 382 (78.3%), 65 (13.3%), and 41 (8.4%) isolates, respectively. Compared to BMD, 96.3% of green AES reports could be confidently and rapidly auto-released, enabling rapid adjustments to antimicrobial therapy. In addition, 69.2% of yellow reports were acceptable, and recommendations to address current AES limitations were made.

**IMPORTANCE** Antimicrobial susceptibility testing (AST) reports are one of the most important clinical microbiology laboratory tasks. AST reports are essential to drive antimicrobial therapy, provide information to monitor antimicrobial resistance rates, and trigger further tests to detect outbreaks or confirm new mechanisms of resistance. Commercial AST devices are frequently used to generate AST reports, and an advanced expert system (AES), such as the Vitek 2 AES, incorporates extensive knowledge to recognize certain susceptibility patterns as indicative of specific phenotypes. Moreover, the Vitek 2 AES also provides a level of confidence for auto-releasing the reports. In this study, the performance of the Vitek 2 AES was compared to state-of-the-art methodologies for AST, broth microdilution and $\beta$-lactamase gene detection, whole-genome sequencing, against a collection of 513 *Enterobacterales* clinical isolates harboring various $\beta$-lactamase genes, including carbapenemase, ESBL, and transferrable AmpC genes, from 73 medical centers in 7 countries in North and Latin America.

**KEYWORDS** ESBL, carbapenemase, AmpC, $\beta$-lactamase, *E. coli*, *K. pneumoniae*, *Escherichia coli*, *Klebsiella pneumoniae*

Generation of antimicrobial susceptibility testing (AST) results is one of the most important roles of the clinical microbiology laboratory to guide antimicrobial therapy and combat the increasing spread of antimicrobial resistance. Most clinical microbiology laboratories use commercial AST (cAST) devices, which are formulated to correlate with the reference broth microdilution (BMD) method. To be cleared by the U.S. FDA for use in clinical laboratories, cAST devices are required to show rates of categorical agreement (CA [i.e., the same susceptible, intermediate, or resistant interpretation]) and essential agreement (EA [i.e., MICs within

Address correspondence to Cecilia G. Carvalhaes, cecilia-carvalhaes@jmilabs.com.

The authors declare a conflict of interest. This study was funded by bioMérieux (Durham, USA). bioMérieux was involved in the study design and decision to present these results and JMI Laboratories received compensation fees for services in relation to preparing the manuscript. bioMérieux was not involved in the collection, analysis, and interpretation of data. JMI Laboratories contracted to perform services in 2019–2020 for Achaogen, Inc., Albany College of Pharmacy and Health Sciences, Allecra Therapeutics, Allergan, AmpliPhi Biosciences Corp., Amicrobe Advanced Biomaterials, Amplyx, Antabio, American Proficiency Institute, Arietis Corp., Arixa Pharmaceuticals, Inc., Astellas Pharma, Inc., Athelas, Basilea Pharmaceutica, Ltd., Bayer AG, Becton, Dickinson and Company, bioMerieux SA, Boston Pharmaceuticals, Bugworks Research, Inc., CEM-102 Pharmaceuticals, Cepheid, Cidara Therapeutics, Inc., CorMedix, Inc., DePuy Synthes, Destiny Pharma, Discuva, Ltd., Dr. Falk Pharma GmbH, Emery Pharma, Entasis Therapeutics, Eurofarma Laboratorios SA, US Food and Drug Administration, Fox Chase Chemical Diversity Center, Inc., Gateway Pharmaceutical LLC, GenePOC, Inc., Geom Therapeutics, Inc., GlaxoSmithKline plc, Harvard University, Helperby, HiMedia Laboratories, F. Hoffmann-La Roche, Ltd., ICON plc, Idorsia Pharmaceuticals, Ltd., Iterum Therapeutics plc, Laboratory Specialists, Inc., Melinta Therapeutics, Inc., Merck & Co., Inc., Microchem Laboratory, Micromyx, MicuRx Pharmaceuticals, Inc., Mutabilis Co., Nabriva Therapeutics plc, NAEJA-RGM, Novartis AG, Oxoid, Ltd., Paratek Pharmaceuticals, Inc., Pfizer, Inc., Polyphor,

*(Continued on next page)*

**TABLE 1** Distribution of *Enterobacterales* species/groups according to the β-lactamase genotype profile

| Molecular category | No. of isolates in category | | | | |
| --- | --- | --- | --- | --- | --- |
| | *E. coli* | *K. pneumoniae* | *E. cloacae* species complex | Other *Enterobacterales* | Total |
| ESBL | 52 | 49 | 11 | 10 | 122 |
| CTX-M-15 + OXA-1 | 13 | 14 | 5 | 5 | 37 |
| CTX-M-15 without other ESBLs | 13 | 6 | 0 | 0 | 19 |
| CTX-M-27 | 18 | 0 | 0 | 1 | 19 |
| ESBL + loss of 1 or 2 porins | 0 | 6 | 0 | 0 | 6 |
| Less frequent ESBLs | 5 | 4 | 0 | 2 | 11 |
| SHV ESBL | 3 | 19 | 6 | 2 | 30 |
| Carbapenemase | 32 | 84 | 45 | 51 | 212 |
| KPC-2 | 5 | 18 | 11 | 12 | 46 |
| KPC-3 | 10 | 20 | 14 | 10 | 54 |
| Other serine-carbapenemase | 2 | 0 | 9 | 18 | 29 |
| Metallo-β-lactamase | 13 | 25 | 10 | 10 | 58 |
| OXA-48-variants | 2 | 15 | 1 | 1 | 19 |
| Double carbapenemases | 0 | 6 | 0 | 0 | 6 |
| tAmpC[a] | 16 | 9 | 1 | 5 | 31 |
| Wild type | 34 | 35 | 15 | 64 | 148 |
| Total | 134 | 177 | 72 | 130 | 513 |

[a]tAmpC, transferrable AmpC.

Ltd., Pharmaceutical Product Development, LLC, Prokaryotics, Inc., Qpex Biopharma, Inc., Roivant Sciences, Ltd., Safeguard Biosystems, Scynexis, Inc., SeLux Diagnostics, Inc., Shionogi and Co., Ltd., SinSa Labs, Spero Therapeutics, Summit Pharmaceuticals International Corp., Synlogic, T2 Biosystems, Inc., Taisho Pharmaceutical Co., Ltd., TenNor Therapeutics, Ltd., Tetraphase Pharmaceuticals, Theravance Biopharma, University of Colorado, University of Southern California-San Diego, University of North Texas Health Science Center, VenatoRx Pharmaceuticals, Inc., Viosera Therapeutics, Vyome Therapeutics, Inc., Wockhardt, Yukon Pharmaceuticals, Inc., Zai Lab, Zavante Therapeutics, Inc. There are no speakers' bureaus or stock options to declare.

±1 $\log_2$ dilution of the reference BMD result]) of ≥90.0% (1, 2). However, the performance levels of cAST devices are not reevaluated when new resistance mechanisms are recognized. Accordingly, many combinations of organisms and antimicrobial agents tested on cAST devices by laboratories today were cleared before many resistance mechanisms were widespread, especially in Gram-negative organisms (3).

The Vitek 2 Advanced Expert System (AES) is a cAST device and system that incorporates extensive knowledge to recognize certain susceptibility patterns as indicative of specific phenotypes and interpret the results accordingly. The Vitek 2 AES validates AST-generated data by checking each MIC value against a database of phenotypes and MIC distributions to infer resistance mechanisms (1, 4). In addition, the system provides a level of confidence for auto-releasing the reports by labeling them either as green (consistent), yellow (consistent with correction), or red (inconsistent, MIC pattern not matching any phenotype). This study evaluates the performance of Vitek 2 AES confidence level dispositions as a tool for the rapid release of AST reports, focusing on β-lactam agents for a set of *Enterobacterales* clinical isolates from North and Latin American medical centers that display a variety of acquired β-lactamases. In addition, the Vitek 2 AES reports were monitored for the ability to produce accurate AST results (EA and CA) compared to broth microdilution (BMD) against this contemporaneous and challenging collection.

## RESULTS

The distribution of *Enterobacterales* species/groups included in this study according to their β-lactamase genotype profile is displayed in Table 1. Further information on the number of isolates per *Enterobacterales* species is available in the supplemental material.

**Overall performance of the Vitek 2 AES system in reporting MIC values and interpretations compared to BMD.** Compared to BMD, the Vitek 2 EA (±1 log dilution) rate was 94.5% and the CA rate was 91.8% when 14,058 MIC values were evaluated against 513 *Enterobacterales* isolates. Among this collection, 365 (71.2%) isolates were further characterized into three molecular groups according to the acquired β-lactamase genes detected: either carbapenemase (212 isolates [41.1%]), extended-spectrum β-lactamase (ESBL) (122 isolates [23.8%]), or transferable AmpC (tAmpC) (31 isolates [6.2%]) (Table 1). A total of 148 (28.8%) isolates were considered wild type (WT): i.e., these isolates did not carry any acquired β-lactamase resistance genes within the categories listed above.

**TABLE 2** Vitek 2 AES essential and categorical agreement rates for *Enterobacterales* isolate subsets compared to the reference BMD method

| Subset (no. of isolates) | No. (%) of isolates with[a]: | | | | |
| | Agreement | | Error | | |
| | EA | CA | mE | ME | VME |
|---|---|---|---|---|---|
| Amoxicillin-clavulanic acid (501) | 486 (97.0) | 454 (90.6) | 40 (8.0) | 2 (1.4) | 5 (1.6) |
| Ampicillin (361) | 359 (99.4) | 354 (98.1) | 7 (1.9) | 0 (0.0) | 0 (0.0) |
| Ampicillin-sulbactam (361) | 354 (98.1) | 329 (91.1) | 27 (7.5) | 2 (2.6) | 3 (1.2) |
| Piperacillin-tazobactam (476) | 441 (92.6) | 431 (90.6) | 38 (8.0) | 5 (2.2) | 2 (0.9) |
| Ceftazidime-avibactam (507) | 460 (90.7) | 500 (98.6) | NA | 2 (0.5) | 4 (6.2) |
| Ceftolozane-tazobactam (462) | 417 (90.3) | 420 (90.9) | 29 (6.3) | 12 (5.1) | 1 (0.5) |
| Cefazolin (513) | 501 (97.7) | 490 (95.5) | NA | 3 (3.3) | 6 (1.4) |
| Cefoxitin (513) | 488 (95.1) | 446 (86.9) | 61 (11.9) | 2 (1.1) | 4 (1.5) |
| Cefuroxime (513) | 491 (95.7) | 492 (95.9) | 20 (3.9) | 0 (0.0) | 1 (0.3) |
| Cefepime (507) | 421 (83.0) | 443 (87.4) | 46 (9.1) | 2 (1.0) | 16 (5.8) |
| Cefotaxime (512) | 469 (91.6) | 500 (97.7) | 7 (1.4) | 1 (0.7) | 4 (1.1) |
| Cefpodoxime (472) | 463 (98.1) | 458 (97.0) | 12 (2.5) | 0 (0.0) | 2 (0.6) |
| Ceftriaxone (435) | 414 (95.2) | 423 (97.2) | 7 (1.6) | 4 (2.8) | 1 (0.4) |
| Aztreonam (513) | 479 (93.4) | 462 (90.1) | 32 (6.2) | 1 (0.6) | 18 (5.6) |
| Ertapenem (476) | 460 (96.6) | 457 (96.0) | 14 (2.9) | 2 (0.8) | 3 (1.4) |
| Imipenem (437) | 400 (91.5) | 407 (93.1) | 28 (6.4) | 2 (0.8) | 0 (0.0) |
| Meropenem (513) | 453 (88.3) | 483 (94.2) | 21 (4.1) | 6 (2.0) | 3 (1.5) |
| Meropenem-vaborbactam (497) | 471 (94.8) | 476 (95.8) | 10 (2.0) | 9 (2.0) | 2 (4.1) |
| Amikacin (494) | 471 (95.3) | 445 (90.1) | 47 (9.5) | 0 (0.0) | 2 (5.1) |
| Gentamicin (480) | 470 (97.9) | 458 (95.4) | 18 (3.8) | 3 (0.9) | 1 (0.8) |
| Tobramycin (475) | 465 (97.9) | 420 (88.4) | 52 (11.0) | 2 (0.7) | 1 (0.6) |
| Ciprofloxacin (474) | 445 (93.9) | 434 (91.6) | 37 (7.8) | 1 (0.5) | 2 (0.8) |
| Levofloxacin (513) | 494 (96.3) | 459 (89.5) | 53 (10.3) | 1 (0.5) | 0 (0.0) |
| Doxycycline (513) | 486 (94.7) | 416 (81.1) | 91 (17.7) | 3 (1.1) | 3 (1.8) |
| Minocycline (513) | 485 (94.5) | 428 (83.4) | 77 (15.0) | 7 (2.0) | 1 (1.0) |
| Tetracycline (513) | 498 (97.1) | 464 (90.5) | 43 (8.4) | 2 (0.8) | 4 (1.7) |
| Tigecycline (488) | 446 (91.4) | 436 (89.3) | 52 (10.7) | 0 (0.0) | 0 (0.0) |
| Nitrofurantoin (513) | 506 (98.6) | 429 (83.6) | 82 (16.0) | 1 (0.6) | 1 (0.4) |
| Trimethoprim-sulfamethoxazole (513) | 490 (95.5) | 491 (95.7) | NA | 5 (2.1) | 17 (6.1) |

[a]EA, essential agreement; CA, categorical agreement; mE, minor error; ME, major error; VME, very major error; NA, not applicable (no intermediary breakpoint).

All $\beta$-lactam agents displayed EA rates of ≥90%, except for meropenem (88.3%) and cefepime (83.0%) (Table 2). The meropenem EA rates were 76.8%, 94.3%, 96.9%, and 98.0% for the carbapenemase, ESBL, tAmpC, and WT subsets, respectively (Table 3). The meropenem CA rate was 94.2% overall, and the rates were 89.6%, 95.1%, 96.9%, and 99.3% for the carbapenemase, ESBL, tAmpC, and WT subsets, respectively. Meropenem category discrepancies among *Enterobacterales* isolates carrying carbapenemase genes occurred mainly due to minor errors (mEs) (16 isolates [7.6%]). Meropenem tested against the carbapenemase subset displayed only 1 (0.6%) very major error (VME) (1 *Serratia marcescens* isolate carrying NDM-1), and 5 major errors (MEs), mainly in isolates exhibiting KPC variants, such as KPC-3 (1 isolate), KPC-4 (2 isolates), and KPC-6 (1 isolate). The remaining isolate displaying meropenem ME was a *Klebsiella oxytoca* isolate carrying the $bla_{VIM-23}$ gene.

The cefepime EA rates for the carbapenemase, ESBL, tAmpC, and WT subsets were 78.8%, 73.0%, 90.6%, and 95.9%, respectively (Table 3). Cefepime exhibited an 87.4% CA rate overall and 83.2%, 82.0%, 90.6%, and 97.2% rates against the carbapenemase, ESBL, tAmpC, and WT subsets, respectively. Minor errors were frequently observed in both categories, carbapenemase (20 occurrences [9.6%]) and ESBL (19 occurrences [15.6%]). In the ESBL subset, cefepime displayed 2 (2.3%) VMEs and 1 (5.9%) ME. The majority of VMEs for cefepime occurred in carbapenemase subset—there were 14 (8.3%) occurrences of VMEs in the carbapenemase subset, while only 1 (3.2%) ME was observed in this group.

Besides cefepime, cefoxitin was the only $\beta$-lactam agent displaying a CA rate of <90%, and category discrepancies occurred mainly due to mEs (61 out of a total of 67 errors [11.9%]) (Table 2). However, elevated VME rates (>1.5%) overall were noted for the following $\beta$-lactam

**TABLE 3** Vitek2 AES essential and categorical agreement rates for resistant *Enterobacterales* subsets

| Subset (no. of isolates) | No. (%) of isolates with[a]: | | | | |
| --- | --- | --- | --- | --- | --- |
| | Agreement | | Error | | |
| | EA | CA | mE | ME | VME |
| **Carbapenemase** | | | | | |
| Amoxicillin-clavulanic acid (206) | 206 (100.0) | 203 (98.5) | 3 (1.5) | 0 (0.0) | 0 (0.0) |
| Ampicillin (128) | 128 (100.0) | 128 (100.0) | 0 (0.0) | NC | 0 (0.0) |
| Ampicillin-sulbactam (128) | 128 (100.0) | 127 (99.2) | 1 (0.8) | 0 (0.0) | 0 (0.0) |
| Piperacillin-tazobactam (186) | 181 (97.3) | 172 (92.5) | 11 (5.9) | 1 (14.3) | 2 (1.2) |
| Ceftazidime-avibactam (208) | 182 (87.5) | 202 (97.1) | NA | 2 (1.4) | 4 (6.7) |
| Ceftolozane-tazobactam (179) | 165 (92.2) | 165 (92.2) | 12 (6.7) | 1 (11.1) | 1 (0.6) |
| Cefazolin (211) | 209 (99.1) | 210 (99.5) | NA | 1 (50.0) | 0 (0.0) |
| Cefoxitin (211) | 204 (96.7) | 180 (85.3) | 30 (14.2) | 0 (0.0) | 1 (0.6) |
| Cefuroxime (211) | 199 (94.3) | 206 (97.6) | 4 (1.9) | 0 (0.0) | 0 (0.0) |
| Cefepime (208) | 164 (78.8) | 173 (83.2) | 20 (9.6) | 1 (3.2) | 14 (8.3) |
| Cefotaxime (210) | 186 (88.6) | 206 (98.1) | 0 (0.0) | 1 (5.3) | 3 (1.6) |
| Cefpodoxime (185) | 181 (97.8) | 182 (98.4) | 2 (1.1) | 0 (0.0) | 1 (0.6) |
| Ceftriaxone (164) | 159 (97.0) | 161 (98.2) | 2 (1.2) | 0 (0.0) | 1 (0.7) |
| Aztreonam (211) | 199 (94.3) | 197 (93.4) | 11 (5.2) | 0 (0.0) | 3 (1.6) |
| Ertapenem (193) | 182 (94.3) | 181 (93.8) | 7 (3.6) | 2 (33.3) | 3 (1.6) |
| Imipenem (174) | 154 (88.5) | 152 (87.4) | 20 (11.5) | 2 (16.7) | 0 (0.0) |
| Meropenem (211) | 162 (76.8) | 189 (89.6) | 16 (7.6) | 5 (25.0) | 1 (0.6) |
| Meropenem-vaborbactam (197) | 179 (90.9) | 182 (92.4) | 8 (4.1) | 5 (3.5) | 2 (4.4) |
| | | | | | |
| **ESBL** | | | | | |
| Amoxicillin-clavulanic acid (122) | 119 (97.5) | 94 (77.1) | 27 (22.1) | 1 (2.2) | 0 (0.0) |
| Ampicillin (105) | 105 (100.0) | 105 (100.0) | 0 (0.0) | NC | 0 (0.0) |
| Ampicillin-sulbactam (105) | 101 (96.2) | 92 (87.6) | 10 (9.5) | 1 (5.0) | 2 (2.8) |
| Piperacillin-tazobactam (122) | 102 (83.6) | 105 (86.1) | 15 (12.3) | 2 (2.6) | 0 (0.0) |
| Ceftazidime-avibactam (122) | 110 (90.2) | 122 (100.0) | 0 (0.0) | 0 (0.0) | 0 (0.0) |
| Ceftolozane-tazobactam (122) | 103 (84.4) | 108 (88.5) | 6 (4.9) | 8 (9.3) | 0 (0.0) |
| Cefazolin (122) | 116 (95.1) | 116 (95.1) | NA | 1 (50.0) | 5 (4.2) |
| Cefoxitin (122) | 113 (92.6) | 101 (82.8) | 20 (16.4) | 1 (1.5) | 0 (0.0) |
| Cefuroxime (122) | 118 (96.7) | 114 (93.4) | 6 (4.9) | 0 (0.0) | 1 (0.9) |
| Cefepime (122) | 89 (73.0) | 100 (82.0) | 19 (15.6) | 1 (5.9) | 2 (2.3) |
| Cefotaxime (122) | 114 (93.4) | 117 (95.9) | 5 (4.1) | 0 (0.0) | 0 (0.0) |
| Cefpodoxime (122) | 121 (99.2) | 118 (96.7) | 4 (3.3) | 0 (0.0) | 0 (0.0) |
| Ceftriaxome (111) | 100 (90.1) | 104 (93.7) | 4 (3.6) | 3 (60.0) | 0 (0.0) |
| Aztreonam (122) | 111 (91.0) | 100 (82.0) | 13 (10.7) | 1 (10.0) | 8 (8.1) |
| Ertapenem (113) | 110 (97.3) | 108 (95.6) | 5 (4.4) | 0 (0.0) | 0 (0.0) |
| Imipenem (122) | 118 (96.7) | 118 (96.7) | 4 (3.3) | 0 (0.0) | 0 (0.0) |
| Meropenem (122) | 115 (94.3) | 116 (95.1) | 3 (2.5) | 1 (1.0) | 2 (11.1) |
| Meropenem-vaborbactam (120) | 114 (95.0) | 116 (96.7) | 2 (1.7) | 2 (1.7) | 0 (0.0) |
| | | | | | |
| **tAmpC[b]** | | | | | |
| Amoxicillin-clavulanic acid (32) | 30 (93.8) | 29 (90.6) | 1 (3.1) | 1 (100.0) | 1 (3.3) |
| Ampicillin (26) | 26 (100.0) | 26 (100.0) | 0 (0.0) | NC | 0 (0.0) |
| Ampicillin-sulbactam (26) | 26 (100.0) | 23 (88.5) | 3 (11.5) | NC | 0 (0.0) |
| Piperacillin-tazobactam (31) | 26 (83.9) | 24 (77.4) | 6 (19.4) | 1 (4.8) | 0 (0.0) |
| Ceftazidime-avibactam (32) | 31 (96.9) | 32 (100.0) | 0 (0.0) | 0 (0.0) | NC |
| Ceftolozane-tazobactam (31) | 25 (80.6) | 23 (74.2) | 8 (25.8) | 0 (0.0) | 0 (0.0) |
| Cefazolin (32) | 32 (100.0) | 32 (100.0) | 0 (0.0) | NC | 0 (0.0) |
| Cefoxitin (32) | 30 (93.8) | 29 (90.6) | 2 (6.3) | 0 (0.0) | 1 (3.3) |
| Cefuroxime (32) | 32 (100.0) | 29 (90.6) | 3 (9.4) | 0 (0.0) | 0 (0.0) |
| Cefepime (32) | 29 (90.6) | 29 (90.6) | 3 (9.4) | 0 (0.0) | 0 (0.0) |
| Cefotaxime (32) | 27 (84.4) | 31 (96.9) | 1 (3.1) | 0 (0.0) | 0 (0.0) |
| Cefpodoxime (31) | 31 (100.0) | 30 (96.8) | 1 (3.2) | 0 (0.0) | 0 (0.0) |
| Ceftriaxone (30) | 26 (86.7) | 28 (93.3) | 1 (3.3) | 1 (16.7) | 0 (0.0) |
| Aztreonam (32) | 23 (71.9) | 20 (62.5) | 6 (18.8) | 0 (0.0) | 6 (40.0) |
| Ertapenem (29) | 28 (96.6) | 28 (96.6) | 1 (3.5) | 0 (0.0) | 0 (0.0) |
| Imipenem (31) | 27 (87.1) | 28 (90.3) | 3 (9.7) | 0 (0.0) | 0 (0.0) |

**TABLE 3** (Continued)

| Subset (no. of isolates) | No. (%) of isolates with[a]: | | | | |
| | Agreement | | Error | | |
| | EA | CA | mE | ME | VME |
|---|---|---|---|---|---|
| Meropenem (32) | 31 (96.9) | 31 (96.9) | 1 (3.1) | 0 (0.0) | 0 (0.0) |
| Meropenem-vaborbactam (32) | 31 (96.9) | 31 (96.9) | 0 (0.0) | 1 (3.1) | NC |
| | | | | | |
| Wild type | | | | | |
| Amoxicillin-clavulanic acid (141) | 131 (92.9) | 128 (90.8) | 9 (6.4) | 0 (0.0) | 4 (9.8) |
| Ampicillin (102) | 100 (98.0) | 95 (93.1) | 7 (6.9) | 0 (0.0) | 0 (0.0) |
| Ampicillin-sulbactam (102) | 99 (97.1) | 87 (85.3) | 13 (12.8) | 1 (1.8) | 1 (3.7) |
| Piperacillin-tazobactam (137) | 132 (96.4) | 130 (94.9) | 6 (4.4) | 1 (0.8) | 0 (0.0) |
| Ceftazidime-avibactam (145) | 138 (95.2) | 145 (100.0) | 0 (0.0) | 0 (0.0) | NC |
| Ceftolozane-tazobactam (130) | 124 (95.4) | 124 (95.4) | 3 (2.3) | 3 (2.5) | 0 (0.0) |
| Cefazolin (148) | 144 (97.3) | 132 (89.2) | 0 (0.0) | 1 (1.1) | 1 (1.7) |
| Cefoxitin (148) | 141 (95.3) | 136 (91.9) | 9 (6.1) | 1 (1.0) | 2 (5.9) |
| Cefuroxime (148) | 142 (95.9) | 130 (87.8) | 7 (4.7) | 0 (0.0) | 0 (0.0) |
| Cefepime (145) | 139 (95.9) | 141 (97.2) | 4 (2.8) | 0 (0.0) | 0 (0.0) |
| Cefotaxime (148) | 142 (95.9) | 146 (98.7) | 1 (0.7) | 0 (0.0) | 1 (4.0) |
| Cefpodoxime (134) | 130 (97.0) | 128 (95.5) | 5 (3.7) | 0 (0.0) | 1 (4.2) |
| Ceftriaxone (130) | 129 (99.2) | 130 (100.0) | 0 (0.0) | 0 (0.0) | 0 (0.0) |
| Aztreonam (148) | 146 (98.6) | 145 (98.0) | 2 (1.4) | 0 (0.0) | 1 (4.8) |
| Ertapenem (141) | 140 (99.3) | 140 (99.3) | 1 (0.7) | 0 (0.0) | 0 (0.0) |
| Imipenem (110) | 101 (91.8) | 109 (99.1) | 1 (0.9) | 0 (0.0) | 0 (0.0) |
| Meropenem (148) | 145 (98.0) | 147 (99.3) | 1 (0.7) | 0 (0.0) | 0 (0.0) |
| Meropenem-vaborbactam (148) | 147 (99.3) | 147 (99.3) | 0 (0.0) | 1 (0.7) | NC |

[a]EA, essential agreement; CA, categorical agreement; mE, minor error; ME, major error; VME, very major error; NA, not applicable (no intermediary breakpoint); NC, not calculable (denominator null).
[b]tAmpC, transferrable AmpC.

agents: amoxicillin-clavulanic acid (1.6% [5 occurrences]), ceftazidime-avibactam (6.2% [4 occurrences]), cefepime (5.8% [16 occurrences as described above]), aztreonam (5.6% [18 occurrences]), and meropenem-vaborbactam (4.1% [2 occurrences]) (Table 2). Ceftolozane-tazobactam (5.1% [12 occurrences]) and cefazolin (3.3% [3 occurrences]) were the only $\beta$-lactam agents exhibiting ME rates of >3.0% in general.

All non $\beta$-lactam agents displayed EA rates of ≥90% against the entire collection of *Enterobacterales*. CA rates were ≥90% for amikacin (90.1%), gentamicin (95.4%), ciprofloxacin (91.6%), tetracycline (90.5%), and trimethoprim-sulfamethoxazole (95.7%) (Table 2). Although tobramycin, levofloxacin, doxycycline, minocycline, tigecycline, and nitrofurantoin exhibited CA rates that ranged from 81.1% to 89.5%, the MEs and VMEs were within acceptable ranges, except for doxycycline (1.8% VMEs in 3 occurrences). Even though CA rates were ≥90%, elevated VME rates were also observed for amikacin (5.1% [2 occurrences]), tetracycline (1.7% [4 occurrences]), and trimethoprim-sulfamethoxazole (6.1% [17 occurrences]).

**AES assessment for rapid AST report.** Among the 513 *Enterobacterales* isolates, 488 had their Vitek 2 AES labeling report evaluated. Twenty-five isolates were excluded from the AES evaluation due to their lack of molecular results. The AES provided reports of consistent (green), consistent with correction (yellow), or inconsistent, MIC pattern not matching any phenotype (red), for 382 (78.3%), 65 (13.3%), and 41 (8.4%) isolates, respectively.

**Consistent (green) reports.** Of all 382 *Enterobacterales* isolates with AES green reports, 167 (43.7%) harbored carbapenemase, 86 (22.5%) ESBL, and 16 (4.2%) tAmpC genes. Additionally, 113 (29.6%) wild-type isolates had a green report issued by the Vitek 2 AES. For 320 (83.8%) isolates, the susceptibility category was correct or only mEs were noted for the $\beta$-lactam antimicrobial agents tested. Table 4 lists all $\beta$-lactam antimicrobials within green reports that displayed at least 1 ME or VME. VME rates of >1.5% were observed for aztreonam (3.5% [8/230 resistant isolates]), cefepime (5.4% [11/204]), ceftazidime-avibactam (1.9% [1/54]), ceftolozane-tazobactam (2.4% [4/168]), and meropenem-vaborbactam (5.8% [3/52]). However, 6 aztreonam VMEs could be corrected by the AES if the aztreonam interpretation was modified to resistant when a high-level cephalosporinase (AmpC) phenotype was detected by the Vitek 2 AES. This improvement would decrease the aztreonam VME rate to

**TABLE 4** VME and ME rates for $\beta$-lactam antimicrobials in *Enterobacterales* green reports by Vitek2 AES

| Antimicrobial agent | No. of isolates[a] | | No. (%) of isolates with error[b,c] | |
|---|---|---|---|---|
| | **Resistant** | **Susceptible** | **ME** | **VME** |
| Amoxicillin-clavulanic acid | 238 | 122 | 2 (1.6) | 1 (0.4) |
| Ampicillin-sulbactam | 259 | 84 | 1 (1.2) | 3 (1.2) |
| Aztreonam | 230 | 130 | 4 (3.1) | 8 **(3.5)** |
| Cefazolin | 303 | 62 | 0 (0.0) | 4 (1.3) |
| Cefepime | 204 | 158 | 2 (1.3) | 11 **(5.4)** |
| Cefotaxime | 252 | 128 | 1 (0.8) | 3 (1.2) |
| Cefoxitin | 207 | 142 | 1 (0.7) | 1 (0.5) |
| Cefpodoxime | 257 | 117 | 0 (0.0) | 1 (0.4) |
| Ceftazidime-avibactam | 54 | 328 | 1 (0.3) | 1 **(1.9)** |
| Ceftolozane-tazobactam | 168 | 206 | 8 **(3.9)** | 4 **(2.4)** |
| Ceftriaxone | 248 | 133 | 3 (2.3) | 1 (0.4) |
| Cefuroxime | 276 | 95 | 7 **(7.4)** | 0 (0.0) |
| Ertapenem | 178 | 200 | 1 (0.5) | 2 (1.1) |
| Imipenem | 171 | 193 | 1 (0.5) | 0 (0.0) |
| Meropenem | 170 | 206 | 3 (1.5) | 1 (0.6) |
| Meropenem-vaborbactam | 52 | 325 | 9 (2.8) | 3 **(5.8)** |
| Piperacillin-tazobactam | 174 | 198 | 2 (1.0) | 2 (1.1) |

[a]Shown are the numbers of isolates resistant or susceptible by broth microdilution.
[b]Error rates of >1.5% for VME and >3.0% for ME are highlighted in boldface.
[c]ME, major error; VME, very major error.

0.9% (2/230). Similarly, the cefepime VME rate would drop from 5.4% to 1.5% if the cefepime category was modified from susceptible to resistant when a carbapenemase phenotype was detected by the Vitek 2 AES. Moreover, the same rule applied to piperacillin-tazobactam would correct the only 2 VMEs observed for this antimicrobial agent in green reports. Notably, 3 of 4 ceftolozane-tazobactam VMEs were detected in *Enterobacterales* isolates expressing carbapenemase phenotypes. Therefore, the ceftolozane-tazobactam VMEs could also be corrected by the AES system, and that would reduce the VME rate from 2.4% to 0.6%. ME rates of >3.0% were only noted for ceftolozane-tazobactam (3.9% [8/206 susceptible isolates]) and cefuroxime (7.4% [7/95]). Although the meropenem-vaborbactam ME rate was 2.8%, for 1 of 9 isolates the susceptibility category could be corrected by the AES if it applied a rule that meropenem-vaborbactam should be reported as susceptible if meropenem alone is susceptible. This rule would reduce the ME rate to 2.5%. Among the 382 green reports, 368 (96.3%) were correctly labeled as green and could be reported with no further evaluation needed.

**Consistent with correction (yellow) reports.** The AES issued 65 reports with yellow labels for 24 isolates that displayed a carbapenemase genotype, 21 with ESBL, 12 with tAmpC, and 8 wild type to acquired $\beta$-lactamase genes. Compared to BMD, at least one of the biological corrections proposed was accepted for 45 (69.2%) isolates, indicating that a yellow report was considered adequate. Four MEs were observed: one for each of the antimicrobials ceftriaxone, meropenem, piperacillin-tazobactam, and ceftolozane-tazobactam (Table 5). VMEs were observed for cefepime (3 events), ceftazidime-avibactam (3 events), cefpodoxime (1 event), cefuroxime (1 event), ertapenem (1 event), and piperacillin-tazobactam (1 event). All 3 cefepime VMEs were noted in *Escherichia coli* (2 events) and *Citrobacter freundii* (1 event) isolates that displayed carbapenemase phenotypes by AES and were confirmed to harbor KPC-2-, KPC-3-, or KPC-4-encoding genes. Similarly, the single cefpodoxime VME occurred in a *Klebsiella pneumoniae* isolate carrying a KPC-3-encoding gene. Improvements to AES may prevent such VMEs from occurring when a carbapenemase phenotype is detected. The 3 ceftazidime-avibactam VMEs occurred in 2 VIM-1-producing *K. pneumoniae* isolates and 1 VIM-23-producing *Klebsiella oxytoca* isolate. Importantly, since carbapenemase phenotypes were reported by AES for those isolates, therapeutic use of ceftazidime-avibactam would be the most appropriate course of action after metallo-$\beta$-lactamase (MBL)-producing strains are excluded or ceftazidime-avibactam susceptibility is confirmed. Among all yellow-labeled

**TABLE 5** VME and ME rates for $\beta$-lactam antimicrobials in *Enterobacterales* yellow reports by Vitek 2 AES

| Antimicrobial agent | No. of isolates[a] | | No. (%) of isolates with error[b] | |
|---|---|---|---|---|
| | Resistant | Susceptible | ME | VME |
| Cefepime | 27 | 30 | 0 (0.0) | 3 (11.1) |
| Cefpodoxime | 53 | 11 | 0 (0.0) | 1 (1.9) |
| Ceftazidime-avibactam | 10 | 55 | 0 (0.0) | 3 (30.0) |
| Ceftolozane-tazobactam | 28 | 32 | 1 (3.1) | 0 (0.0) |
| Ceftriaxone | 49 | 15 | 1 (6.7) | 0 (0.0) |
| Cefuroxime | 51 | 9 | 0 (0.0) | 1 (2.0) |
| Ertapenem | 20 | 38 | 0 (0.0) | 1 (5.0) |
| Meropenem | 15 | 44 | 1 (2.3) | 0 (0.0) |
| Piperacillin-tazobactam | 31 | 29 | 1 (3.4) | 1 (3.2) |

[a]Shown are the numbers of isolates resistant or susceptible by broth microdilution.
[b]ME, major error; VME, very major error.

reports, 16 (24.6%) isolates were in agreement with BMD results and could be labeled green and reported with no further evaluation needed.

**Inconsistent (red) reports.** Red reports were issued for 41 (8.4%) of 488 isolates evaluated. Among these 41 isolates, 20, 15, and 4 isolates carried carbapenemase-, ESBL-, and/or tAmpC-encoding genes, respectively. No acquired $\beta$-lactamase genes were detected in 2 isolates (wild type). A total of 33 (80.5%) isolates displayed Vitek 2 MIC interpretations concordant with the BMD method (no ME or VME). In addition, although a carbapenemase phenotype was not released by the AES, meropenem resistance was detected by Vitek 2 in 13 of 20 isolates that displayed a carbapenemase genotype. The 7 remaining isolates carried $bla_{KPC}$ (1 KPC-4-producing *E. coli* isolate and 1 KPC-2-producing *Proteus mirabilis* isolate) or $bla_{OXA-48}$ variant genes (2 *E. coli* isolates carrying $bla_{OXA-181}$, 2 *K. pneumoniae* isolates carrying $bla_{OXA-163}$, and 1 *Enterobacter cloacae* complex isolate harboring $bla_{OXA-163}$). Notably, these isolates were susceptible to all carbapenems tested by BMD and Vitek 2 and so they potentially expressed low levels of carbapenemase or exhibited a poor hydrolytic profile. Of the $\beta$-lactam antimicrobials reported by AES in red reports, only 4 MEs were observed—one for each of the agents ertapenem, imipenem, meropenem, and meropenem-vaborbactam (Table 6). Moreover, 5 VMEs were also noted—2 for cefepime and 1 for each of the $\beta$-lactam antimicrobials aztreonam, ceftriaxone, and ertapenem. The 2 cefepime VMEs occurred in isolates harboring carbapenemase genes and displaying a meropenem-resistant phenotype by the AES. Notably, red labels are applied by AES due to a technical problem or when the organism is expressing a phenotype that is not present in the AES knowledge base. Therefore, if phenotypes are not identified during AES assessment, the microbiologist should consider additional testing, organism reidentification, and/or retesting AST before reporting.

**TABLE 6** VME and ME rates for $\beta$-lactam antimicrobials in *Enterobacterales* red reports by Vitek 2 AES

| Antimicrobial agent | No. of isolates[a] | | No. (%) of isolates with error[b] | |
|---|---|---|---|---|
| | Resistant | Susceptible | ME | VME |
| Aztreonam | 32 | 9 | 0 (0.0) | 1 (3.1) |
| Cefepime | 30 | 9 | 0 (0.0) | 2 (6.7) |
| Ceftriaxone | 32 | 6 | 0 (0.0) | 1 (3.1) |
| Ertapenem | 18 | 21 | 1 (4.8) | 1 (5.6) |
| Imipenem | 11 | 24 | 1 (4.2) | 0 (0.0) |
| Meropenem | 11 | 26 | 1 (3.8) | 0 (0.0) |
| Meropenem-vaborbactam | 2 | 37 | 1 (2.7) | 0 (0.0) |

[a]Shown are the numbers of isolates resistant or susceptible by broth microdilution.
[b]ME, major error; VME, very major error.

## DISCUSSION

Antimicrobial resistance is a rising concern worldwide. In 2019, the U.S. Centers for Disease Control and Prevention declared that more than 2.8 million antibiotic-resistant infections occur in the United States each year, and more than 35,000 people die as a result (5). Carbapenem-resistant *Enterobacterales* and extended-spectrum $\beta$-lactamase (ESBL)-producing *Enterobacterales* are among the pathogens recognized as urgent and serious threats, respectively. It is recognized that patients infected with resistant pathogens are more likely to receive ineffective empirical antibiotic therapy, which is associated with poor outcomes, including death (6–10). On the other hand, the use of broad-spectrum therapies for the empirical treatment of serious infections may promote further selection of antimicrobial resistance, increased toxicity, and higher costs of care (11, 12). In this respect, AST could provide a prediction of the likelihood of an antimicrobial's clinical success and allow the emergence and spread of resistant microorganisms to be monitored (2).

The Vitek 2 device is widely used in clinical laboratories worldwide, providing the advantage of standardizing endpoint reading and often producing rapid susceptibility test results in 3.5 to 16 h (13). In addition, the AES enhances the ability to analyze test results for atypical patterns and unusual resistance phenotypes (13). Vitek 2 AES uses the susceptibility testing results to infer resistance mechanisms by comparing its results to a large database and correcting susceptibility interpretations, assisting the microbiologists and clinicians in patient care decision-making (4, 14). In this study, we challenged the Vitek 2 AES with a large collection of contemporaneous clinical *Enterobacterales* isolates from North and Latin America that harbored clinically important resistance genes and rigorously evaluated MIC values and interpretations against the reference BMD method (4). Here, the majority of the *Enterobacterales* isolates (71.2%) harbored carbapenemase, ESBL, or tAmpC genes, and only 28.8% were wild type for these mechanisms of $\beta$-lactam resistance. In the over 14,058 pathogen/antimicrobial combinations tested, Vitek 2 displayed EA and CA rates of $\geq$90.0%. Although the CLSI recommendations for EA, CA, and error rates were taken into consideration when evaluating this challenging collection of *Enterobacterales*, it is important to emphasize that these criteria were established for testing isolates routinely encountered in clinical laboratories.

In the evaluation of each $\beta$-lactam antimicrobial, CA rates of <90% were noted for cefepime (87.4%) and cefoxitin (86.9%), mainly due to mEs. In addition, cefepime VMEs occurred mainly in isolates harboring carbapenemase genes (14/16). The cefepime VME rate could be reduced from 5.8% to 1.8% by improving AES corrections based on the organism phenotype, such as modifying the cefepime susceptibility category from susceptible to resistant when a carbapenemase phenotype is recognized.

The Vitek 2 AES identifies a phenotype to three confidence levels: (i) green, indicating consistent or typical, where all MICs match with the phenotype; (ii) yellow, indicating consistent with correction or atypical, where one MIC does not match with the closest phenotype (s); and (iii) red, indicating inconsistent, where at least two MICs do not match with any phenotype or a phenotype cannot be identified with enough confidence. All atypical and inconsistent reports were reviewed and verified by the microbiologist (14). The ability to use the AES green report as a rapid tool for susceptibility testing was also assessed. In this challenging collection of *Enterobacterales* isolates, all green (consistent) reports were evaluated by an experienced microbiologist and the MIC values and interpretations were compared to BMD results. Overall, 368 (96.3%) of 382 susceptibility reports would be released without further tests or modification. This assessment is in accordance with a previous study that evaluated Vitek 2 routine reports, where 99.3% of typical reports were confirmed as consistent (14). However, in our study, for 14 isolates, the microbiologist review or further improvements to AES would potentially prevent VMEs for aztreonam, cefepime, ceftolozane-tazobactam, and piperacillin-tazobactam and MEs for meropenem-vaborbactam. Notably, in our study, cefepime results by Vitek 2 trended toward lower MIC values than BMD, an observation previously noted by Cantón and colleagues when evaluating Vitek 2 for ESBL detection (15). These trends were mainly noted with TEM- and SHV-type ESBLs, but in our collection they mostly occurred in the carbapenemase subset and mainly in *Enterobacterales* isolates carrying KPC-type enzymes.

In the ESBL subset, this trend was noted with CTX-M-15- and CTX-M-27-type ESBLs and, less frequently, with SHV.

The assessment of AES reports labeled with a confidence level of yellow (consistent with corrections) resulted in 16 reports in agreement with BMD results, and these could be released as consistent with no further evaluation. Moreover, for 35 reports, the corrections proposed by the AES were considered adequate compared to BMD and an experienced microbiologist review. However, for 5 of 12 remaining isolates, cefepime, cefpodoxime, and piperacillin-tazobactam VMEs could be detected by an experienced microbiologist upon review and additional tests or by improving the Vitek 2 AES. Similarly, the study conducted by Pages Monteiro and colleagues noted that in 68% of the yellow reports for routine Gram-positive and Gram-negative bacterial isolates included in the study, at least 1 MIC result was not confirmed by additional tests (Etest or disc diffusion) and in 32% of the yellow reports, the susceptibility result could be released without further tests (14).

Interestingly, in AES reports labeled red (inconsistent); the susceptibility report was consistent with the BMD method in 80.5% of isolates. Moreover, of great clinical importance, carbapenem resistance was reported for 16 isolates harboring carbapenemase genes, regardless of the AES carbapenemase phenotype. For all red reports, a microbiologist review is required along with confirmation of purity and identification. In addition, alternative confirmatory susceptibility tests may be required.

In summary, this study showed that the AES confidence level report is a valuable tool for clinical laboratories. *Enterobacterales* green susceptibility reports can be rapidly released, allowing for prompt adjustments to antimicrobial use if these results are quickly communicated to an antimicrobial stewardship team. Although the AES dispositions of yellow reports were adequate for most of the isolates in this category, review by an experienced microbiologist is advised since further confirmatory tests may be required. Red or inconsistent reports will require a microbiologist to review for an additional check on the purity and identification and may result in additional laboratory tests. Moreover, based on reference methods—BMD and whole-genome sequencing—suggestions for improving AES could be made. Furthermore, because newer mechanisms of resistance could arise and currently identified ones could become more predominant, microbiologists must be vigilant as to how well each test method can accurately detect resistance. Our results showed that the Vitek 2 AES continues to provide accurate susceptibility testing for contemporaneous *Enterobacterales* isolates harboring diverse mechanisms of resistance to $\beta$-lactams.

## MATERIALS AND METHODS

***Enterobacterales* isolates.** A total of 513 *Enterobacterales* clinical isolates from North and Latin America were tested, representing 73 medical centers, including 62 medical centers in the United States (9 U.S. Census divisions, 390 isolates [76.0% overall]) and 11 medical centers in Latin America (6 countries, 123 isolates [24.0% overall]) (see the supplemental material). Of those isolates, 407 were recovered from documented infections during 2015 to 2019 and selected from the SENTRY Antimicrobial Surveillance Program. A further 106 isolates were selected from the Centers for Disease Control and Prevention and the U.S. FDA Antimicrobial Resistant Isolate Bank (AR Isolate Bank) based on their $\beta$-lactamase profile. Only a single isolate per patient was included. Bacterial isolates were identified by standard microbiology methods and matrix-assisted laser desorption ionization–time of flight mass spectrometry (MALDI-TOF MS) (Bruker Daltonics, Bremen, Germany).

**Antimicrobial susceptibility testing.** Isolates were tested for antimicrobial susceptibility by BMD following the guidelines in CLSI document M07-A11 (16) and by the Vitek 2 system using N802 and XN15 Vitek 2 test cards following the manufacturer's recommendations. The Vitek 2 MIC results were generated using the AES in the Global Clinical and Laboratory Studies Institute (CLSI)-based + Natural Resistance (NATR) mode. The AES report for each isolate and card was reviewed by an experienced microbiologist. The following antimicrobial agents were included in this study: amikacin, amoxicillin-clavulanic acid, ampicillin, ampicillin-sulbactam, aztreonam, cefazolin, cefepime, cefotaxime, cefoxitin, cefpodoxime, ceftazidime-avibactam, ceftolozane-tazobactam, ceftriaxone, cefuroxime, ciprofloxacin, doxycycline, ertapenem, gentamicin, imipenem, levofloxacin, meropenem, meropenem-vaborbactam, minocycline, nitrofurantoin, piperacillin-tazobactam, tetracycline, tigecycline, tobramycin, and trimethoprim-sulfamethoxazole. Susceptibility interpretations published by CLSI (17) were applied where available. For cefazolin, the CLSI breakpoint for oral administration was applied, and the U.S. FDA breakpoint for tigecycline against *Enterobacterales* was used.

Vitek 2 and BMD MIC values were validated by concurrently testing the following ATCC quality control (QC) reference strains: *E. coli* ATCC 25922 and ATCC 35218, *K. pneumoniae* ATCC 700603 and ATCC BAA-2814, and *Pseudomonas aeruginosa* ATCC 27853. The inoculum density during susceptibility testing was monitored by bacterial colony counts or according to the manufacturer's instructions. Where applicable, QC ranges for tested reference strains were those published in CLSI M100 (17) or provided by the manufacturer.

**Molecular characterization.** *Enterobacterales* isolates displaying meropenem- and/or imipenem-elevated MIC results of >1 mg/L (imipenem was not applied to *Proteus mirabilis* or indole-positive *Proteus*) and/or *E. coli* and *Klebsiella pneumoniae* isolates displaying MIC values of ≥2 mg/L for at least 2 of the $\beta$-lactams, aztreonam, cefepime, ceftazidime, and/or ceftriaxone, were submitted to molecular characterization of $\beta$-lactamase-encoding genes by whole-genome sequencing as previously described (18). Isolates were then classified into the following $\beta$-lactam-resistant genotypes: carbapenemase, extended spectrum $\beta$-lactamase (ESBL), and transferrable (plasmid) AmpC. Importantly, the ESBL group only included isolates harboring ESBL genes that did not coharbor carbapenemase genes. Similarly, the transferrable AmpC group included isolates that harbored transferrable AmpC genes but did not coharbor ESBL or carbapenemase genes. The carbapenemase group included isolates that harbored carbapenemase genes regardless of the presence of ESBL or transferrable AmpC genes. *Enterobacterales* isolates that did not meet the criteria for molecular characterization were considered wild type.

**Essential and categorical agreement assessment.** EA was considered when MIC values obtained with the reference method and Vitek 2 system were identical, within ±1 log dilution, or in potential agreement, with off-scale MIC values that could potentially be equivalent to or within ±1 log dilution of each other. The CA rate was assessed when the Vitek 2 and reference method susceptibility category were equivalent to CLSI criteria applied (17). The CLSI criteria were used to calculate VME, ME, and mE interpretation errors between the Vitek 2 system and the reference method (BMD). Vitek 2 false-susceptible interpretations were considered VMEs and calculated based upon the number of resistant isolates, while false-resistant interpretations were considered MEs and calculated based upon the number of susceptible isolates (19). mEs occurred when one method reported a result as intermediate and the other method reported the result as susceptible or resistant. CLSI recommends EA and CA rates of ≥90% to verify each combination of antimicrobial agent and microorganism against a collection of organisms routinely isolated in the clinical laboratory and VME and ME rates of ≤1.5% and ≤3.0%, respectively (19).

**AES dispositions and assessment.** The AES dispositions (levels of confidence of green, yellow, and red) were compared to BMD results for accuracy. AES green reports were assessed for the number (percentage) of isolates that were correctly classified and accepted to be automatically posted. The AES yellow reports displayed recommendations for editing the susceptibility category and/or the MIC value to match the detected phenotype. AES red reports were technical errors or on organisms that expressed a phenotype that was not present in the knowledge base and therefore would require additional testing.

## SUPPLEMENTAL MATERIAL

Supplemental material is available online only.

**SUPPLEMENTAL FILE 1**, PDF file, 0.1 MB.

## ACKNOWLEDGMENTS

We thank all participants of the SENTRY Surveillance Program for providing bacterial isolates. We also would like to thank Amy Chen and Maria Morabe for editorial assistance.

Cecilia Carvalhaes contributed Conceptualization, Methodology, Validation, Formal Analysis, Data Curation, Supervision, Writing – Original Draft; Dee Shortridge contributed Investigation, Methodology, Writing – Review & Editing; Leah N. Woosley contributed Methodology, Validation, Formal Analysis, Data Curation, Supervision; Nabina Gurund contributed Methodology, Testing, Validation; Mariana Castanheira contributed Methodology, Resources, Writing – Review & Editing, Funding Acquisition.

This study was funded by bioMérieux (Durham, NC, USA). bioMérieux was involved in the study design and decision to present these results, and JMI Laboratories received compensation fees for services in relation to preparing the manuscript. bioMérieux was not involved in the collection, analysis, and interpretation of data.

JMI Laboratories contracted to perform services in 2019 to 2020 for Achaogen, Inc., Albany College of Pharmacy and Health Sciences, Allecra Therapeutics, Allergan, AmpliPhi Biosciences Corp., Amicrobe Advanced Biomaterials, Amplyx, Antabio, American Proficiency Institute, Arietis Corp., Arixa Pharmaceuticals, Inc., Astellas Pharma, Inc., Athelas, Basilea Pharmaceutica, Ltd., Bayer AG, Becton Dickinson and Company, bioMérieux SA, Boston Pharmaceuticals, Bugworks Research, Inc., CEM-102 Pharmaceuticals, Cepheid, Cidara Therapeutics, Inc., CorMedix, Inc., DePuy Synthes, Destiny Pharma, Discuva, Ltd., Dr. Falk Pharma, GmbH, Emery Pharma, Entasis Therapeutics, Eurofarma Laboratorios SA, U.S. Food and Drug Administration, Fox Chase Chemical Diversity Center, Inc., Gateway Pharmaceutical, LLC, GenePOC, Inc., Geom Therapeutics, Inc., GlaxoSmithKline, plc, Harvard University, Helperby, HiMedia Laboratories, F. Hoffmann-La Roche, Ltd., ICON, plc, Idorsia Pharmaceuticals, Ltd., Iterum Therapeutics plc, Laboratory Specialists, Inc., Melinta Therapeutics, Inc., Merck & Co., Inc., Microchem Laboratory, Micromyx, MicuRx Pharmaceuticals, Inc., Mutabilis Co., Nabriva Therapeutics, plc, NAEJA-RGM, Novartis AG, Oxoid, Ltd., Paratek Pharmaceuticals, Inc., Pfizer,

Inc., Polyphor, Ltd., Pharmaceutical Product Development, LLC, Prokaryotics, Inc., Qpex Biopharma, Inc., Roivant Sciences, Ltd., Safeguard Biosystems, Scynexis, Inc., SeLux Diagnostics, Inc., Shionogi and Co., Ltd., SinSa Labs, Spero Therapeutics, Summit Pharmaceuticals International Corp., Synlogic, T2 Biosystems, Inc., Taisho Pharmaceutical Co., Ltd., TenNor Therapeutics, Ltd., Tetraphase Pharmaceuticals, Theravance Biopharma, University of Colorado, University of Southern California—San Diego, University of North Texas Health Science Center, VenatoRx Pharmaceuticals, Inc., Viosera Therapeutics, Vyome Therapeutics, Inc., Wockhardt, Yukon Pharmaceuticals, Inc., Zai Lab, and Zavante Therapeutics, Inc. There are no speakers' bureaus or stock options to declare.

We declare no conflict of interest.

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
