## [Reviewer comments · Microbiology Spectrum]

Microbiology Spectrum

Performance of the VITEK 2 Advanced Expert System (AES) as a Rapid Tool for Reporting Antimicrobial Susceptibility Testing (AST) in Enterobacterales from North and Latin America

Cecilia Carvalhaes, Dee Shortridge, Leah Woosley, Nabina Gurung, and Mariana Castanheira

Corresponding Author(s): Cecilia Carvalhaes, JMI Laboratories

Review Timeline:

Submission Date:

November 15, 2022

Accepted:

December 20, 2022

Editor: Paul Luethy

Reviewer(s): Disclosure of reviewer identity is with reference to reviewer comments included in decision letter(s). The following individuals involved in review of your submission have agreed to reveal their identity: Brian Mochon (Reviewer #2)

Transaction Report:

DOI: <https://doi.org/10.1128/spectrum.04673-22>

December 20, 2022

Dr. Cecilia G Carvalhaes
JMI Laboratories
North Liberty, IA 52317

Re: Spectrum04673-22 (Performance of the VITEK 2 Advanced Expert System (AES) as a Rapid Tool for Reporting Antimicrobial Susceptibility Testing (AST) in Enterobacterales from North and Latin America)

Dear Dr. Cecilia G Carvalhaes:

Your manuscript has been accepted, and I am forwarding it to the ASM Journals Department for publication. You will be notified when your proofs are ready to be viewed.

Sincerely,

Paul Luethy
Editor, Microbiology Spectrum

Journals Department
Performance of the VITEK 2 Advanced Expert System (AES) as a Rapid Tool for Reporting Antimicrobial susceptibility Testing (AST) in Enterobacterales from North and Latin America

A comprehensive evaluation of the performance of the VITEK 2 AES confidence level dispositions as well as accuracy compared to BMD. Furthermore, it provides an updated look at this particular ES system performance since the emergence of various resistance mechanisms using an interesting mix of isolates. This study characterizes and categorizes isolates with elevated MICs by their resistance genotypes obtained by WGS. This paper only focused on B-lactam drugs for Enterobacterales isolates displaying variety of B lactamases. The paper was clear and straightforward and data heavy consisting primarily of tables to display the data in a digestible format. The main weakness is addressed by the author in that comparison are largely done using CLSI CA, EA and error rates recommendations in the evaluation, despite the recommendations being established for routinely encountered isolates in a clinical setting. Otherwise, a nice comprehensive update of the performance of the AES system in question against some challenging isolates. Criticism based really on the labeling of tables.

Table 2 – line 438- has the abbreviation for not calculable (NC), but this is not in Table 2.

Need consistency with nomenclature rules- some places “Enterobacterales” is italicized but not always.

Not sure why Tables 4-6 have abbreviation key for ME and VME when it was used in earlier tables that did not have this key.